# POINTS: Improving Your Vision-language Model with Affordable Strategies

## Abstract

In recent years, vision-language models have achieved significant advancements, excelling in tasks once deemed challenging, such as optical character recognition and geometric problem-solving. Despite these impressive achievements, several critical issues remain unaddressed: 1) Proprietary models rarely disclose detailed information about their architectures. In contrast, while open-source models provide visibility into their training strategies, detailed ablations of these strategies are highly anticipated. 2) Pre-training data is currently under-explored in open-source works, with most efforts empirically adding datasets from diverse sources, making the entire process elusive and cumbersome. 3) During the fine-tuning stage, the focus is often on adding and ablating more datasets, which frequently leads to diminishing returns. Therefore, refining data schemes is essential for further enhancing model performance. To address these issues, we propose the following contributions in this paper: 1) We trained a robust baseline model, leveraging the latest technological advancements in vision-language models. Building upon existing advancements, we introduced effective improvements and conducted comprehensive ablation and validation for each technique incorporated into this strong baseline. 2) Inspired by recent work on large language models, we propose filtering pre-training data using perplexity, selecting the data with the lowest perplexity as the training set. This approach allowed us to train on a curated 1M dataset, resulting in highly competitive performance. 3) During the visual instruction tuning stage, we experimented with model soup on different datasets when further introducing more datasets into the training set brought marginal improvements. Integrating these innovations, we obtained a model with 9B parameters, performing competitively with a series of existing state-of-the-art models. Additionally, these strategies we propose are efficient and relatively lightweight, allowing the community to adopt them easily for their models.

## 1 Introduction

Advancements in large language models (LLMs; Chowdhery et al. 2023, Jiang et al. 2023, OpenAI 2022, Yang et al. 2024, Dubey et al. 2024) have significantly enhanced the capabilities of vision-language large models (Fu et al. 2023, Liu et al. 2023b, OpenAI 2023, Dong et al. 2024a, Zhu et al. 2023), enabling more sophisticated analyses of textual and visual information. Prominent closed-source model paradigms such as GPT-4 (OpenAI, 2023), Gemini Pro 1.5 (Fu et al., 2023), and Claude 3 (Anthropic, 2024) have achieved remarkable success in expanding LLMs into the realm of vision-language models. Concurrently, open-source vision-language large models are also advancing rapidly, with numerous notable contributions emerging in the field (Liu et al., 2024b; Chen et al., 2024d).

Historically, LLaVA (Liu et al., 2024b) has served as a common baseline. However, recent advancements have rendered its performance suboptimal. Thus, there is a need to establish a stronger baseline for further exploration. In this work, we enhance the vanilla LLaVA architecture by refining the pre-training dataset. Inspired by CapFusion (Yu et al., 2024), we merge the original captions with world knowledge and generated captions that exhibit good grammatical structure. For visual instruction tuning datasets, we introduce Individual Select (Liu et al., 2024c) to curate effective instruction tuning datasets. Regarding model architecture, we first incorporate Dynamic High Resolution to help the model capture fine-grained details. To address image distortion issues inherent

in Dynamic High Resolution, we propose a novel image splitting strategy called Consistent Aspect Ratio Dynamic High Resolution (CATTY), which maintains a consistent image ratio. Additionally, inspired by Vary (Wei et al., 2023), we merge features from a vision encoder trained separately with text-rich data with those from the original vision encoder, significantly boosting the model's optical character recognition (OCR) capabilities. Unlike most existing works (Li et al., 2024a; Chen et al., 2024d), we extensively ablate each newly introduced component in the strong baseline to verify their individual benefits.

Recent works seldom explore the optimization of pre-training datasets. Most studies (Chen et al., 2024d; Yao et al., 2024; Bai et al., 2023b) tend to empirically combine samples from various large-scale datasets (Schuhmann et al., 2022; Byeon et al., 2022), often leading to inefficient and computationally expensive pre-training processes. In the domain of large language models, some research leverages perplexity to filter pre-training datasets. Inspired by this approach, we filter our pre-training dataset by selecting the top samples with the lowest perplexity values. This filtering process yields a subset of 1 million data samples, on which we subsequently pre-train our model. Experimental results demonstrate that the model trained on this filtered subset outperforms a model trained on a dataset five times larger.

In the visual instruction tuning stage, most existing works (Liu et al., 2024c; Li et al., 2024a; Chen et al., 2024d) focus on collecting large quantities of datasets and performing ablation studies to select the most effective ones. However, this approach often reaches a plateau, where introducing additional datasets yields only marginal or even degraded performance. Previous research on model soup has demonstrated the benefits of merging weights from different models fine-tuned with various hyper-parameters. In this work, we propose using model soup to merge weights from models fine-tuned with different datasets to further improve performance when dataset selection no longer brings significant improvement. Compared to conducting model soup on models fine-tuned with different hyper-parameters, *e.g.* learning rate, the improvement with model soup on models fine-tuned with different datasets is much more prominent. Following this line of work, we further experiment with different model soup strategies and find that greedy model soup is the most effective.

By integrating the aforementioned innovations, we have developed a model called **POINTS**. Our contributions are threefold:

- We propose a strong baseline that integrates the latest advancements in vision-language models and thoroughly verify the effectiveness of each component.
- We introduce the use of perplexity to filter the pre-training dataset and conduct a detailed investigation of data distribution across different perplexity intervals.
- We employ model soup to merge models fine-tuned with different datasets, thereby enhancing model performance when further dataset selection yields only marginal improvements.

## 2 RELATED WORKS

**Multimodal Large Language Models** The rapid advancement of large language models (LLMs; Dubey et al. 2024, Team et al. 2023, Achiam et al. 2023, Yang et al. 2024, Su et al. 2022) has laid the groundwork for the emergence of multimodal large language models (MLLMs; Li et al. 2024a, Liu et al. 2024b, Liu et al. 2024a, Bai et al. 2023a, Qiao et al. 2024), which aim to integrate visual understanding with language reasoning and multimodal perception and comprehension. Prominent models such as GPT-4v (Achiam et al., 2023) and Gemini-1.5-Pro (Team et al., 2023), developed by major corporations, have spearheaded the MLLM era, utilizing proprietary training data and undisclosed training methodologies. Meanwhile, open-source models have been striving to keep pace. For instance, LLaVA-Next (Liu et al., 2024a) and InternVL-1.5 (Chen et al., 2024d) introduce dynamic high-resolution techniques by dividing a large image into multiple smaller segments with ratio-inconsistent resizing. MiniCPM-V (Yao et al., 2024) employs a specialized vision encoder to generate non-square image patches. Additionally, models like Vary (Wei et al., 2023), SPHINX (Lin et al., 2023), Cambrian-1 (Tong et al., 2024), and Mini-Gemini (Li et al., 2023a) propose dual vision encoders to enhance visual capabilities. Furthermore, the significant progress in multimodal model evaluation (Liu et al., 2023c; Chen et al., 2024c; Fang et al., 2024) has also contributed to the rapid improvement of large vision-language models. In this work, we introduce POINT, a model trained

exclusively with fully open-source datasets during both the pre-training and supervised fine-tuning (SFT) stages, demonstrating promising results on extensive benchmarks.

**Visual Instruction Tuning**   The selection of training data for multimodal models is of paramount importance (Laurençon et al., 2024; Tong et al., 2024), and most improvements in existing works stem from detailed ablation of instruction tuning datasets (Li et al., 2024a; Liu et al., 2024b; Chen et al., 2024d). The commonly used approach to select the most effective datasets involves iteratively adding each dataset to the pool; if it brings improvement, we keep it, otherwise, we drop it. However, this approach may eventually plateau, as further additions might only yield marginal improvements. Previous workHe et al. (2024) has shown the benefits of weight merging, but their experimental results are relatively preliminary. To further enhance performance, we systematically propose employing model soup (Wortsman et al., 2022) on different models fine-tuned with various visual instruction tuning datasets. This method involves merging the model weights after visual instruction tuning on diverse datasets, resulting in notable performance improvements.

## 3 METHODS

This section is divided into three parts: i) In subsection 3.1, we integrate various techniques from previous methods (Liu et al., 2024a; Lin et al., 2023; Wei et al., 2023; Liu et al., 2024c; Chen et al., 2024d) to create a strong baseline for further experiments. Additionally, we propose a novel dynamic resolution splitting method, termed **C**onsistent **A**spect **R**atio **D**ynamic High Resolution (**CATTY** for short), to mitigate the issue of image distortion. ii) In subsection 3.2, we propose using perplexity to filter the pre-training dataset. iii) Finally, in subsection 3.3, we incorporate the concept of model soup (Wortsman et al., 2022) into the instruction tuning stage. We find that this straightforward approach can significantly improve the model's performance, especially when further data selection only brings marginal or even degraded performance.

### 3.1 A STRONG BASELINE

In this section, we integrate the recent advancements from existing works to create a strong baseline, containing **Dynamic High Resolution** from InternVL1.5(Chen et al., 2024d), **CapFusion** from (Yu et al., 2024), **Dual Vision Encoder** from Vary(Wei et al., 2023) and SPHINX(Lin et al., 2023), **Individual Select** from (Liu et al., 2024c). Following LLaVA(Liu et al., 2024b), POINTS mainly contains three parts: vision encoder, projector and the large language model. By integrating all these practices from previous works, we obtain the model structure and pipeline in Figure 1.

**Dynamic High Resolution**   It has been verified that feeding high-resolution images to vision-language models is beneficial for capturing fine-grained details and reducing hallucinations (Liu et al., 2023b). To enable vision encoder with fixed input resolutions to accommodate dynamic image resolutions, Dynamic High Resolution in LLaVA-Next (Liu et al., 2024a) and InternVL-1.5 (Chen et al., 2024d) splits high-resolution images into several tiles of the same resolution, which the original vision encoder can process. The concrete steps are as follows: i) First, the maximum number of tiles an image can be split into is predefined (set to 8 in our experiments). ii) Based on the maximum number of tiles, a table is created containing information about the target image before splitting. The key of the table is the aspect ratio, and the value is the width and height of the target image, which can be evenly divided by the resolution of the vision encoder. iii) For each image, the target resolution is fetched from the pre-computed table according to the similarity between aspect ratios. The current image is then resized to the target resolution and split into several tiles of the same resolution.

**Consistent Aspect Ratio Dynamic High Resolution (CATTY)**   Before splitting the image, Dynamic High Resolution in InternVL-1.5 (Chen et al., 2024d) resizes the image to the target resolution. However, this resizing is not proportional to the image's original aspect ratio, which can cause distortion. This issue has been discussed in previous articles(Yao et al., 2024). Therefore, we propose a splitting method that maintains the image's aspect ratio, named Consistent Aspect Ratio Dynamic High Resolution (see Figure 2). The first two steps in CATTY are the same as those in InternVL-1.5, and the last step works as follows: Given an image with height H and width W, we obtain the height and width of the referenced image from the pre-computed table, denoted as $H^r$ and $W^r$, respectively. Then, we resize the image to the target size ($H^t \times W^t$) by:

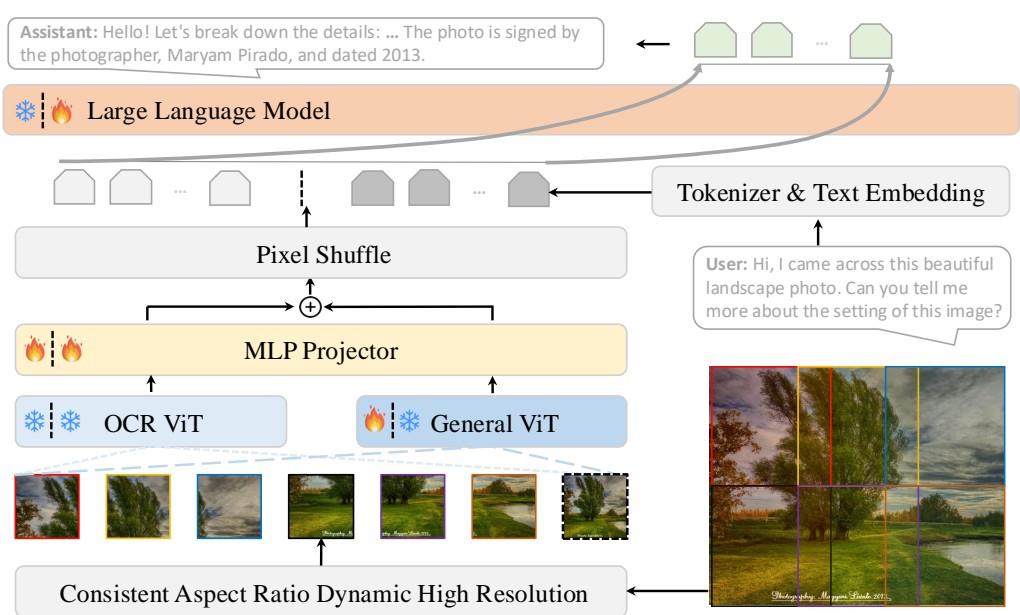

Figure 1: **The architecture of POINTS.** For each module (*e.g.* OCR ViT, General ViT, MLP Projector, and Large language model), the label to the left of the dash line indicates the status during pre-training, while the label to the right indicates the status during the instruction tuning stage.

$$\text{ratio} = \min(H, W)/\min(H^r, W^r)$$
$$H^t = \text{ratio} \times H \tag{1}$$
$$W^r = \text{ratio} \times W$$

Given the input resolution of a vision encoder, $H^v \times W^v$, the target image should be divided into $\frac{H^r}{H^v} \times \frac{W^r}{W^v}$ tiles. Next, we split the target image, $H^t \times W^t$, using a sliding window with strides $(S^h, S^w)$ across the height and width, respectively. The strides $(S^h, S^w)$ are computed as follows:

$$S^h = (H^t - H^v)/(H^r/H^v - 1)$$
$$S^w = (W^t - W^v)/(W^r/W^v - 1) \tag{2}$$

In Equation 2, $S^h$ is set to 0 if $H^r/H^v = 1$, and similarly for $S^w$. This approach allows us to divide a high-resolution image into several tiles without introducing any distortion. There is one exception: if the aspect ratio of the original image is larger than 8, we resize it to an aspect ratio of 1:8 by default. Alongside the tiles obtained using CATTY, we also include a thumbnail of the global view of the image to capture the overall context. This thumbnail is resized to match the input resolution of the vision encoder. Before feeding the features output by the vision encoder into the large language model, we employ the *pixel shuffle* technique with a down-sampling factor of 0.25, as described in InternLM-XComposer2-4KHD (Dong et al., 2024b), to reduce the sequence length of the image features for improved efficiency.

**CapFusion**    The original captions in existing pre-training datasets are often noisy and structurally flawed, making them sub-optimal for model training. To address this, synthetic captions, such as those in LAION-COCO and BLIP-LAION (Li et al., 2022), generated by image captioning models, have been proposed. However, the simplistic syntactic and semantic structures in synthetic captions may contribute to issues like *Scalability Deficiency and World Knowledge Loss* (Yu et al., 2024). CapFusion strikes a balance between these two types of captions by utilizing a large language model to organically integrate raw and synthetic captions. This approach extracts real-world knowledge from the structurally flawed raw captions while merging it with the structured but syntactically simplified synthetic captions. Following the CapFusion methodology, we use InternLM-XComposer2

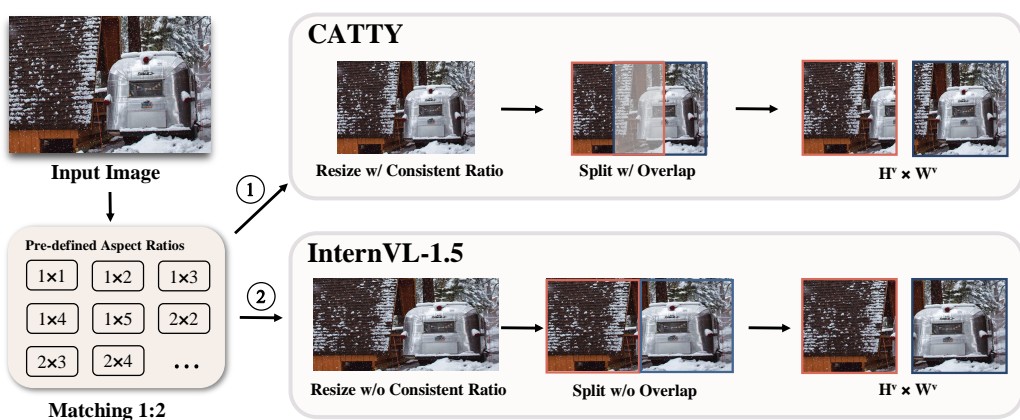

Figure 2: **Comparison between dynamic high resolution in InternVL-1.5 and Consistent Aspect Ratio Dynamic High Resolution (CATTY) proposed by us.**

(Dong et al., 2024a) to generate captions for images and InternLM2 (Cai et al., 2024) to integrate the original raw and synthetic captions. The prompts to generate image captions and merge captions are in the Appendix.

**Dual Vision Encoder** Several previous works, such as SPHINX (Lin et al., 2023) and Cambrian-1 (Tong et al., 2024), have demonstrated that different vision encoders exhibit distinct advantages across various domains. Combining features from multiple encoders can lead to improved and more robust performance. Unlike the perception and reasoning required for natural images, text-intensive images demand different capabilities from vision-language models (Wei et al., 2023). To enhance optical character recognition (OCR) capabilities, we train a separate vision encoder, referred to as the OCR ViT, to extract textual features from images, following the methodology of Vary (Wei et al., 2023). Unlike Vary, we do not construct training samples, such as charts, ourselves; instead, we utilize OCR results (extracted using PaddleOCR in our case) for pre-training. Additionally, we include natural captions in the pre-training dataset for the OCR vision encoder. More details about the composition of the pre-training datasets for the OCR vision encoder will be discussed in the following section. We merge the features from the general vision encoder (General ViT) and the OCR vision encoder using a weighted average before feeding them into the large language model.

**Individual Select** Individual Select, as proposed by Liu et al. (2024c), aims to identify the most effective instruction tuning datasets. Building on this approach, we adopt the dataset composition from Liu et al. (2024c) as our candidate pool and incorporate additional datasets used in DeepSeek-VL (Lu et al., 2024a), Cambrian-1 (Tong et al., 2024), and Cauldron (Laurençon et al., 2024b). Ultimately, we integrate 16 more datasets into those identified by Liu et al. (2024c) (further details are provided in Appendix). To enhance the diversity of prompts, given the homogeneity in the style of prompts within academic datasets, we employ GPT-4o to generate question-answer pairs in line with previous works (Lu et al., 2024a; Chen et al., 2024d) (the prompt to generate question-answer pairs will be provided in the Appendix). The images for these pairs are randomly selected from LAION-5B (Schuhmann et al., 2022). We refer to the final composition of visual instruction tuning datasets as the **Base Set**.

## 3.2 PRE-TRAIN DATA SELECTION

In the context of large language models, perplexity has long been employed as a metric to assess the quality of pre-trained datasets (Albalak et al., 2024; Marion et al., 2023). Inspired by this approach, we utilize an off-the-shelf vision-language model, $P$—either the model obtained through the steps outlined in subsection 3.1 or an open-sourced VLM—to further filter out low-quality pre-trained datasets obtained via Capfusion, as described above. For each item, $s$, in the pre-trained dataset mentioned in subsection 3.1, we compute the perplexity for all text tokens using the following formula:

$$\text{Perplexity}(s) = \exp(-\frac{1}{N} \sum_{i=1}^{N} \log P(w_i|w_1, w_2, ..., w_{i-1})) \tag{3}$$

Let $\{w_1, \ldots, w_N\}$ represent the text token sequence for $s$. We sort all these items in ascending order and select the first 20% for the pre-training stage. Upon closer examination of the first and last 20% of items, we observe that the distinguishing factor is not the quality of the data, which contrasts with observations in large language models. The last 20% of items often contain obscure world knowledge, such as game version numbers and computer factory serial numbers. This type of world knowledge is extremely rare and contains very little information, making it less beneficial for the model's learning. In the Appendix, we provide some examples randomly sampled from the first and last 20% of items.

### 3.3 INSTRUCTION DATA SELECTION WITH MODEL SOUP

Visual instruction tuning data is crucial for the superior performance of existing vision-language models (Chen et al., 2024d; Dong et al., 2024a; Liu et al., 2024b). However, most existing works focus on selecting more effective datasets by iterative ablation. In many cases, this approach reaches a plateau, where further data selection can only bring marginal improvements or even degrade performance. In this section, we systematically introduce the benefits of using model soup to integrate the advantages of models fine-tuned with different instruction tuning datasets after data selection meets a bottleneck. The philosophy behind model soup is as follows: given a pre-trained model, fine-tuning the model with different hyper-parameters, $h_1, \ldots, h_k$, results in several fine-tuned models converging to different local optima, denoted as $f(\theta_1, h_1), \ldots, f(\theta_k, h_k)$. These hyper-parameters include learning rate, data augmentation, initialization seed, etc. By interpolating the weights of these fine-tuned models, we can always obtain a stronger model, $f(\theta_s, h_s)$. Given the pre-trained model obtained through the methods discussed above, a base instruction tuning dataset $D$, and a series of visual instruction tuning datasets $d_1, \ldots, d_k$ to be selected, we can obtain a stronger model using the following steps:

- For each dataset $d_i \in \{d_1, ..., d_k\}$, we add it to the base instruction tuning dataset, $D$, to obtain an augmented dataset, $D_i^*$.
- We train $k$ models using each augmented from $\{D_1^*, ..., D_k^*\}$ concurrently, and obtain $\{f(D_1^*; \theta_1), ..., f(D_k^*; \theta_k)\}$.
- We select $p$ models from $\{f(D_1^*; \theta_1), ..., f(D_k^*; \theta_k)\}$, and merge the weights from all these selected models to obtain a stronger model.

For the third step above, we choose several methods to select the best composition of fine-tuned models to obtain a final model with superior performance, namely, *Maximum Soup*, *Average Soup*, and *Greedy Soup*.

**Maximum Soup**  Given an evaluation score, Acc, we can obtain a strong model, $f(\theta_s)$, using the following formula:

$$\{\theta_i\}_{\text{len}(\{\theta_i\})=p} = \text{Arg}_{(\theta_i)}(\text{Top}_p(\{\text{Acc}(f(D_1^*; \theta_1)), ..., \text{Acc}(f(D_k^*; \theta_k))\})))$$
$$f(\theta_s) = f(\frac{1}{p} \sum_{i=1}^{p} \theta_i) \tag{4}$$

**Average Soup**  By taking the average of weights from all fine-tuned models, we can obtain a stronger model, $f(\theta_s)$:

$$f(\theta_s) = f(\frac{1}{k} \sum_{i=1}^{k} \theta_i) \tag{5}$$

**Greedy Soup**  We start by sorting the fine-tuned models in descending order based on their evaluation scores. Next, we iterate through these sorted models. For each model, we compute the average

of its weights with those of all models currently in the model pool. If the evaluation score improves, the model is added to the pool. Finally, we average the weights of all models in the pool to obtain a stronger model, denoted as $f(\theta_s)$. The table below outlines the detailed pipeline of Greedy Soup.

---

**Algorithm 1** Greedy Soup for Visual Instruction Tuning Datasets

---

1: **INPUT:** $k$ fine-tuned models with different datasets, $\{f(D_i^*; \theta_i)\}$
2: **INPUT:** the evaluation score, Acc
3: **INPUT:** model pool, P $\leftarrow \{\}$
4: **for** $i = 1$ to $k$ **do**
5:    **if** $\text{Acc}(f(\text{average}(P, \theta_i)))) \geq \text{Acc}(f(\text{average}(P))$ **then**
6:       P $\leftarrow$ P $\cup \theta_i$
7:    **end if**
8: **end for**
9: Return average(P)

---

## 4 EXPERIMENTS

This section is divided into five subsections: (i) evaluation setup, (ii) pre-training and instruction-tuning datasets used to train the strong baseline (iii) details about the training setup for the OCR ViT pre-training, the vision-language pre-training, and the visual instruction tuning stages, (iv) ablation studies and analyses of each component used to build our final model, and (v) comparison with other works on extensive benchmarks.

### 4.1 EVALUATION SETUP

Before embarking on our exploration, we sought a robust evaluation metric to comprehensively assess the various capabilities of our model. This is where OpenCompass (Contributors, 2023) proves helpful. OpenCompass proposes eight benchmarks to balance the evaluation of a model from different perspectives. These benchmarks include MMBench (Liu et al., 2023c) and MMStar (Chen et al., 2024b) for diagnosing general abilities, MMMU (Yue et al., 2024) for testing STEM-related abilities, HallusionBench (Liu et al., 2023a) for model hallucination, MathVista (Lu et al., 2023) for math-related abilities, AI2D (Kembhavi et al., 2016) for chart-related abilities, OCRBench (Liu et al., 2023d) for OCR capabilities, and MMVet (Yu et al., 2023b) for subjective evaluation. By averaging the metrics from these benchmarks, OpenCompass derives a score that represents the comprehensive ability of a model. Additionally, it offers a useful tool, VLMEvalKit (Duan et al., 2024), for one-click evaluation. Therefore, unless otherwise specified, we will use these eight benchmarks for our ablation study, with the exception of MMBench, for which we will use the *dev-en* split.

### 4.2 DATA SETUP

**Pre-train Dataset** To train the OCR ViT, we randomly selected 20 million data points from LAION-5B-en (Schuhmann et al., 2022), LAION-5B-cn (Schuhmann et al., 2022), WuKong (Gu et al., 2022), and Zero (Gu et al., 2022). We then used PaddleOCR to extract text from the images, replacing the original captions to form new image-caption pairs for pre-training. Following Vary (Wei et al., 2023), we also included 10 million original data samples from LAION-5B, where the captions are the original ones crawled from the Internet. However, we did not adopt the cumbersome pipeline of constructing a new dataset for OCR enhancement, such as crawling PDF files and converting them to images for training (Bai et al., 2023b), as we found our existing pipeline already performs well on OCR-related tasks. For the vision-language pre-training in constructing the strong baseline, we used CapFusion to construct 20 million data points (note that these data do not overlap with those used in OCR ViT pre-training) from LAION-5B. From this set, we selected 5 million data points, as we found this setting works best, similar to the observation in Liu et al. (2024c). Based on the 5 million data, we further selected a 1 million dataset for the final vision-language alignment by choosing the top 20% of data with the lowest perplexity value.

**Visual Instruction Tuning Dataset** Based on the datasets identified by Liu et al. (2024c), we further employ **Individual Select** to choose additional datasets from those proposed in (Lu et al., 2024a), (Tong et al., 2024), and (Laurençon et al., 2024b). The final composition of datasets, referred to as the **Base Set**, used to construct the robust baseline is presented in Appendix.

## 4.3 TRAINING SETUP

**Pre-training Setup for OCR ViT** The pre-training framework follows the standard LLaVA-style architecture (Liu et al., 2023b), comprising a vision encoder, a two-layer MLP, and a large language model. The vision encoder is initialized from OpenAI's CLIP-ViT-Large-336, while the large language model is initialized from Yi-1.5-9B-Chat (Young et al., 2024). Throughout the pre-training stage, the large language model remains frozen, whereas the vision encoder and MLP are trainable. The learning rates for the vision encoder and MLP are set to $2 \times 10^{-4}$ and $2 \times 10^{-5}$, respectively, with a warm-up schedule during the first 3% of steps, followed by a cosine decay schedule for the remaining steps.

**Setup for the Vision-language Pre-training Stage** The General ViT, depicted in Figure 1, is initialized from OpenAI's CLIP-ViT-Large-336, while the OCR ViT is derived from the preceding stage. For the General ViT, only the last three layers are trainable, as this configuration yielded the best results in our experiments. The OCR ViT remains frozen throughout this stage, consistent with the settings used in Vary(Wei et al., 2023). Features from the penultimate layer of both the General and OCR ViT are selected and fed into the projector. The projector itself is a two-layer MLP, which remains tunable during the pre-training stage. The learning rates for the General ViT and the MLP are set to $2 \times 10^{-4}$ and $2 \times 10^{-5}$, respectively. A warm-up schedule is applied during the first 3% of steps, followed by a cosine decay schedule for the remaining steps.

**Setup for the Visual Instruction Tuning Stage** Both the General ViT and OCR ViT remain frozen throughout the entire stage. The learning rates for the projector and the large language model are both set to $2 \times 10^{-5}$. A warm-up schedule is applied during the first 3% of steps, followed by a cosine decay schedule for the remaining steps.

## 4.4 ABLATION STUDY AND ANALYSIS

**Each Component to Build the Strong Baseline** As shown in Table 1, each component introduced in subsection 3.1 contributes to steady improvements. These enhancements are significant; for instance, after introducing Dynamic High Resolution to split the input image, we observe substantial improvements in OCR-related tasks, such as OCRBench, with performance increasing from 49.6% to 55.6%. Additionally, the use of high-resolution images with Dynamic High Resolution helps reduce hallucination, primarily due to the increased detail in the high-resolution images. Furthermore,

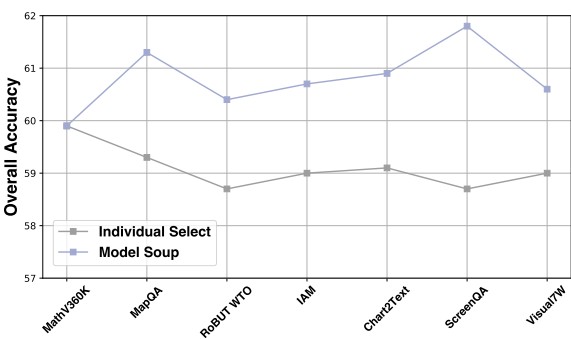

Figure 3: **The superiority of Model Soup.** When adding additional instruction tuning datasets no longer yields benefits (Individual Select), Model Soup can significantly enhance performance.

replacing the original Dynamic High Resolution with CATTY results in notable improvements across various benchmarks, with OCR-related benchmarks showing greater gains than others. This is likely because image distortion has a more pronounced negative impact on text within images. Compared to general visual feature extraction, the ability to extract text features from images is limited for CLIP-ViT(Radford et al., 2021), as it was trained on a large quantity of general image-text pairs. Consequently, we observe substantial improvements on OCRBench after integrating features from an additional ViT, post-trained on text-rich images. Among the 5 strategies, incorporating more visual instruction tuning datasets by Individual Select yields the most significant improvements. This observation aligns with existing works(Chen et al., 2024d; Li et al., 2024a; Tong et al., 2024), underscoring the importance of selecting effective datasets during the visual instruction tuning stage.

| CF | DHR | CATTY | DVE | IS | MMB | MV | HB | OCR | AI2D | MMVet | MMStar | MMMU | Overall |
|----|-----|-------|-----|----|-----|----|----|-----|------|-------|--------|------|---------|
| | | | | | 72.1 | 43.8 | 35.7 | 48.9 | 73.2 | 40.0 | 51.5 | 36.2 | 50.2 |
| ✓ | | | | | 74.8 | 44.8 | 35.5 | 49.6 | 74.5 | 41.2 | 51.8 | 36.4 | 51.1 |
| ✓ | ✓ | | | | 75.1 | 45.1 | 38.0 | 55.6 | 75.0 | 42.2 | 52.6 | 37.5 | 52.6 |
| ✓ | | ✓ | | | 75.9 | 45.7 | 39.1 | 56.9 | 75.8 | 43.1 | 52.4 | 37.9 | 53.4 |
| ✓ | | ✓ | ✓ | | 77.3 | 49.2 | 42.3 | 60.3 | 76.0 | 44.5 | 54.3 | 40.1 | 55.5 |
| ✓ | | ✓ | ✓ | ✓ | **80.1** | **57.4** | **44.2** | **69.2** | **76.4** | **47.2** | **54.5** | **43.3** | **59.0** |

Table 1: **Ablation about each component to build the strong baseline.** CF: CapFusion(Yu et al., 2024), DHR: Dynamic high resolution(Chen et al., 2024d), CATTY: Consistent aspect ratio dynamic high resolution proposed by us, DEV: Dual vision encoder(Wei et al., 2023), IS: Individual select(Liu et al., 2024c). MMB: the *dev-en* split of MMBench(Liu et al., 2023c), MV: MathVista(Lu et al., 2023), HB: HallusionBench(Liu et al., 2023a), OCR: OCRBench(Liu et al., 2023d), Overall: the average of scores on the first 8 benchmarks.

| #num | perplexity | Overall |
|------|-----------|---------|
| 5M | | 59.0 |
| 20M | | 58.8 |
| 1M | ✓ | **59.6** |

| DVE | SVE | OCR | Overall |
|-----|-----|-----|---------|
| ✓ | | **69.2** | **59.0** |
| | ✓ | 67.3 | 57.2 |

| lr | ds | Model Soup | Overall |
|----|----|-----------|---------|
| | | baseline | 59.0 |
| ✓ | | Greedy Soup | 59.2 |
| | ✓ | Maximum Soup | 61.0 |
| | ✓ | Average Soup | 61.2 |
| | ✓ | Greedy Soup | **61.8** |

Table 2: The first two rows compare the use of different data quantities during the pre-training stage. The third row represents a subset of the 5M dataset from the first row, filtered by perplexity.

Table 3: **DVE**: Dual Vision Encode. **SVE**: Single Vision Encoder, incorporating the OCR dataset used to train the OCR ViT into the dataset for the vision-language pre-training stage.

Table 4: Comparison of different model soup strategies over visual instruction tuning datasets. **lr**: model soup on models fine-tuned with different learning rates. **ds**: model soup on models fine-tuned with different datasets.

**Pre-train Dataset** As shown in Table 2, scaling up the dataset size (constructed by CapFusion) from 5M to 20M results in downgraded performance, similar to the observations in Liu et al. (2024c). Additionally, some works also achieve promising performance using relative small pre-training datasets instead of a huge number of datasets during the pre-training stage (Li et al., 2024a; Liu et al., 2024a). We believe the possible reasons are: i) The vision encoder of most existing vision-language models is initialized from a pre-trained model that has already been trained on a large quantity of image-text pairs. It is highly likely that most of the data used in the vision-language pre-training stage has already been seen by the vision encoder, thus bringing only marginal or even negative impact when scaling up the size of the vision-language pre-training dataset. ii) The pre-training datasets are quite homogeneous for existing large-scale web-crawled datasets, *e.g.*, LAION-5B and COYO-700M (Byeon et al., 2022). We plot the distribution of the main entity for each image of a subset extracted from LAION-5B in the Appendix and find that this distribution is long-tailed and constrained to a few objects, *e.g.*, person. Thus, indiscriminately pre-training the model on such datasets can only bring limited benefits. As shown in the third row, we can improve performance by pre-training the model on merely 1M data, coming from the top 20% of the 5M data in the first row, which has the lowest perplexity. This result shows that excessively exposing the model to obscure and scarce knowledge during the transition is detrimental to its learning. Furthermore, compared to fusing features from a separate OCR-enhanced vision encoder, introducing a large OCR dataset has two obvious drawbacks: i) During the pre-training stage, the model has to align both the general features and OCR-related features, which may result in conflicts (Wei et al., 2023). ii) Since the size of the dataset used in vision-language pre-training is relatively small, a large OCR dataset may overwhelm the learning process, which is not helpful for learning other kinds of knowledge. Thus, introducing features from another OCR ViT can yield superior performance in Table 3.

**Improve the Performance with Model Soup on Different Datasets.** As described in previous sections, increasingly adding more instruction tuning datasets often reaches a plateau where further increasing the number of datasets yields minimal improvement. However, by incorporating model soup across different datasets, we observe substantial enhancements, as shown in Table 4, with the overall score increasing from 59.0 to 61.8. We also compare the benefits of various model soup strategies. Among them, greedy soup achieves the best performance, outperforming maximum soup

| Methods | MMB | MV | HB | OCR | AI2D | MMVet | MMStar | MMMU | SCI | MME | RWQ | Wild |
|---|---|---|---|---|---|---|---|---|---|---|---|---|
| *Proprietary models* | | | | | | | | | | | | |
| GPT-4o-0513 | - | 61.3 | 55.0 | 73.6 | 84.6 | 69.1 | 63.9 | 69.2 | 90.7 | 2310.3 | 75.4 | 102.0 |
| Claude3.5-Sonnet | - | 61.6 | 49.9 | 78.8 | 80.2 | 66.0 | 62.2 | 65.9 | 88.9 | 1920.0 | 60.1 | 81.0 |
| Gemini-1.5-Pro | - | 57.7 | 45.6 | 75.4 | 79.1 | 64.0 | 59.1 | 60.6 | 85.7 | 2110.6 | 64.1 | 95.3 |
| *Open-source models* | | | | | | | | | | | | |
| Cambrian-34B | 81.4 | 50.3 | 41.6 | 59.1 | 79.5 | 53.2 | 54.2 | 50.4 | 85.6 | 2049.9 | 67.1 | 82.0 |
| Ovis1.5-LLaMA3-8B | - | 63.0 | 45.0 | 74.4 | 82.5 | 50.9 | 57.3 | 48.3 | 88.8 | 1948.5 | 64.2 | 79.9 |
| Idefics3-LLaMA3-8B | - | 58.4 | 43.7 | 55.0 | 76.5 | 41.7 | 55.0 | 46.6 | 91.3 | 1937.4 | 62.6 | 66.3 |
| InternVL2-8B | - | 58.3 | 45.0 | 79.4 | 83.6 | 54.3 | 61.5 | 51.2 | 97.1 | 2215.1 | 64.2 | 73.3 |
| IXC-2.5 | - | 63.7 | 43.1 | 68.6 | 81.6 | 49.3 | 59.9 | 42.9 | 96.6 | 2233.1 | 67.8 | 70.2 |
| OneVision | 80.8 | 62.3 | 31.6 | 62.2 | 82.4 | 51.9 | 61.9 | 47.9 | 95.4 | 1993.6 | 69.9 | 81.0 |
| *Ours* | | | | | | | | | | | | |
| POINTS-9B | 83.2 | 60.7 | 48.0 | 70.6 | 78.5 | 50.0 | 56.4 | 46.9 | 92.9 | 2017.8 | 65.9 | 69.3 |

Table 5: **Comparison between different methods.** MMB: the *dev-en* split of MMBench(Liu et al., 2023c), MV: MathVista(Lu et al., 2023), HB: HallusionBench(Liu et al., 2023a), OCR: OCRBench(Liu et al., 2023d), SCI: ScienceQA(Lu et al., 2022a), MME: MME(Yin et al., 2023), RWQ: RealWorldQA, Wild: LLaVA-Wild(Liu et al., 2024b). Cambrian-34: Cambrian-34B(Tong et al., 2024), Ovis1.5-LLaMA3-8B: Ovis1.5(Lu et al., 2024b), IXC-2.5: InternLM-XComposer-2.5(Zhang et al., 2024), OneVision: LLaVA-OneVision(Li et al., 2024a), Idefics3-LLaMA3-8B: IDEFICS3 (Laurençon et al., 2024a). The language model POINTS-9B uses is Yi-1.5-9B (Young et al., 2024). Results are obtained from the leaderboard of OpenCompass, except for MMBench. POINTS-7B uses is Qwen-2.5-7B (Team, 2024).

and average soup by 0.8 and 0.6 points, respectively. Unless otherwise specified, we will use greedy soup by default in subsequent experiments. Additionally, we include the results of conducting model soup over different hyperparameters, *e.g.* different learning rates. As shown, model soup over hyperparameters brings only marginal improvement. Furthermore, we verify in the Appendix that model soup consistently improves performance regardless of the **Base Set** used.

### 4.5 COMPARISON WITH OTHER WORKS

In addition to the 8 benchmarks used in the ablation studies above, we further include ScienceQA (Lu et al., 2022a), MME (Yin et al., 2023), LLaVA-Wild (Liu et al., 2024b), and ReadWorldQA to compare the performance of different models. The following table shows the performance of these models. As shown in Table 5, POINTS achieves performance comparable to existing state-of-the-art models of similar size and even surpasses models with much larger sizes, such as Cambrian-34B. Additionally, compared to the models listed in the table, POINTS uses a much smaller pre-training dataset (e.g., 1M), fewer visual instruction tuning datasets, and all the datasets we used are publicly available. This makes it more affordable for the community to adopt the strategies proposed in this paper. Furthermore, each aspect of POINTS is clearly presented and thoroughly analyzed, making the effectiveness of each strategy employed in our model evident.

## 5 CONCLUSION

Vision-language models have achieved significant progress in recent years. Following this trend (Chen et al., 2024d; Li et al., 2024a; Liu et al., 2024b; Zhang et al., 2024; Tong et al., 2024), we first establish a strong baseline by integrating various advancements proposed in recent works (Liu et al., 2024a; Yu et al., 2024; Wei et al., 2023; Liu et al., 2024c) for further experiments. Additionally, we delve into the intricate details of these advancements and propose effective refinements, such as the Consistent Aspect Ratio Dynamic High Resolution. We also conduct extensive experiments to verify the effectiveness of each component in constructing the strong baseline. Secondly, we propose using perplexity to filter the pre-training dataset, retaining only the top 20% of data with the smallest perplexity values during the pre-training stage. This filtering method also brings significant improvements. Model Soup (Wortsman et al., 2022) has shown promising potential to further enhance performance by averaging the weights of fine-tuned models with different hyperparameters. However, we find that conducting model soup over different dataset settings can yield even more substantial improvements.

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

## A  VISUAL INSTRUCTION TUNING DATASETS

Table 6 shows the visual instruction tuning datasets used in this work.

## B  PROMPT FOR CAPFUSION

Figure 4 shows the prompts to generate new image caption and merge the original caption and the generated caption.

| Category | Dataset |
|---|---|
| Conversation | LLaVAR(Zhang et al., 2023), inhouse GPT-4o data, Mini-Gemini(Li et al., 2024b) LVIS-Instruct4V(Wang et al., 2023) |
| Document | DocVQA(en)(Mathew et al., 2021) |
| Caption | ALLaVA(Chen et al., 2024a), ShareGPT4V(Chen et al., 2023), LAION-GPT4V |
| General QA | VSR(Zhang et al., 2021), IConQA(Lu et al., 2021b) |
| Science | AI2D(Kembhavi et al., 2016), TQA(Kim et al., 2018), ScienceQA(Lu et al., 2022a) |
| Chart&Screen | DVQA(Kafle et al., 2018), POIE(Kuang et al., 2023), MapQA(Chang et al., 2022) ScreenQA(Hsiao et al., 2022) |
| Mathematics | GeoQA+(Cao & Xiao, 2022), Geo3K(Lu et al., 2021a), TabMWP(Lu et al., 2022b) CLEVR-Math(Lindström & Abraham, 2022), SuperCLEVER(Li et al., 2023b), MathV360K(Shi et al., 2024) |
| Knowledge | KVQA(Sanket Shah & Talukdar, 2019) |
| OCR | InfoVQA(Mathew et al., 2022), TextVQA(Singh et al., 2019) ST-VQA(Biten et al., 2019), ICDAR2015, HME100K(Yuan et al., 2022) |
| Text-only | LIMA(Zhou et al., 2024), Alpaca-GPT4(Peng et al., 2023) OpenHermes2.5(Teknium, 2023), MetaMathQA(Yu et al., 2023a) MathInstruct(Yue et al., 2023), orca-math-word-problems-200k(Mitra et al., 2024) atlas-math-sets, Math |

Table 6: Visual instruction tuning datasets to build the strong baseline and the those finally selected to conduct model soup (marked in red).

---

## Generate Image Caption

<ImageHere>Please briefly describe the image in English

## Fuse Original Caption and Generated Image Caption

The following two sentences are different descriptions of the same picture, please merge and refine the information in the two given sentences.

Sentence 1 provides detailed world knowledge, but there are defects in sentence structure and grammar. Sentence 2 shows good sentence structure, but lacks in-depth real-world details and may contain erroneous information.

Please merge them into a new sentence, ensuring good sentence structure while retaining the detailed real-world information provided in sentence 1.
There are several requirements:
1.  Please organically combine the descriptions of the two sentences about the picture, without any traces of adhesion.
2.  At the same time, do not introduce any information that has not appeared in these two sentences.
3.  Please only return the merged sentence, do not provide other information.

Sentence 1: {original caption}
Sentence 2: {generated image caption}
Merged Sentence:

---

Figure 4: **Prompt for image caption generation and captions merging.**

## C    PROMPT FOR INHOUSE GPT-4O DATASET

Figure 5 shows the prompt to generate the inhouse GPT-4o data in Table 6.

> ## ## Generate inhouse GPT-4o Dataset
>
> You are an AI visual assistant, please ask two questions about the content of the image and give the corresponding answer.
> Requirements:
> 1. The question need complex reasoning to be answered.
> 2. The answer can be obtained confidently
> 3. Provide detailed answers when answering complex questions. For example, give detailed examples or reasoning steps to make the content more convincing and well-organize
> 4. The perspective of asking questions needs to be more diverse, and at the same time, the sentence structure of the questions also needs to be more varied.
> 5. Please give question&answer pairs in the following format with no serial number:
>    <question>
>    <answer>

Figure 5: **Prompt to generate the inhouse GPT-4o dataset.**

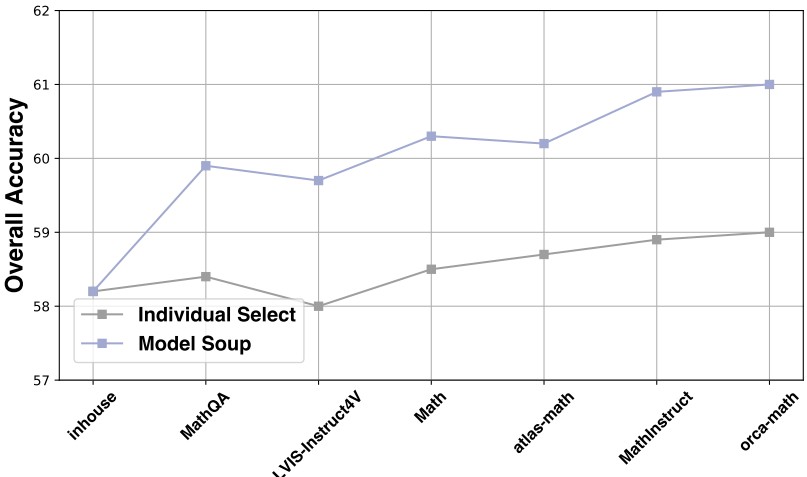

Figure 6: Model Soup brings consistent improvement regardless of what Base Set is used.

## D   MODEL SOUP WITH DIFFERENT BASE SET

To verify that model soup consistently improves performance regardless of the Base Set used, we randomly sampled 6 datasets from the Base Set in Table 6 to conduct model soup, while the remaining datasets were used as the new Base Set. As shown in Figure 6, model soup also brings significant improvements compared to individual selection, demonstrating the effectiveness and universality of model soup.

## E   HOMOGENEITY IN EXISTING PRE-TRAINING DATASET

We randomly sampled 5 million images from LAION-5B and used POINTS to identify the main object in each image. We then plotted the distribution of the top six objects from the sampled data. As shown on the left side of Figure 7, these six objects account for more than 90% of the total data. The right side of Figure 7 illustrates the distribution of the top six objects after applying a simple balancing technique: if the count of a particular object exceeds the average count of the top six objects, we down-sample it to 60% of its total count. We re-trained the strong baseline model on both the original and balanced pre-training datasets. The overall score of the balanced version outperformed the original by 0.6. This is an initial investigation into the distribution of the pre-training dataset, and we plan to explore this direction further in our future research.

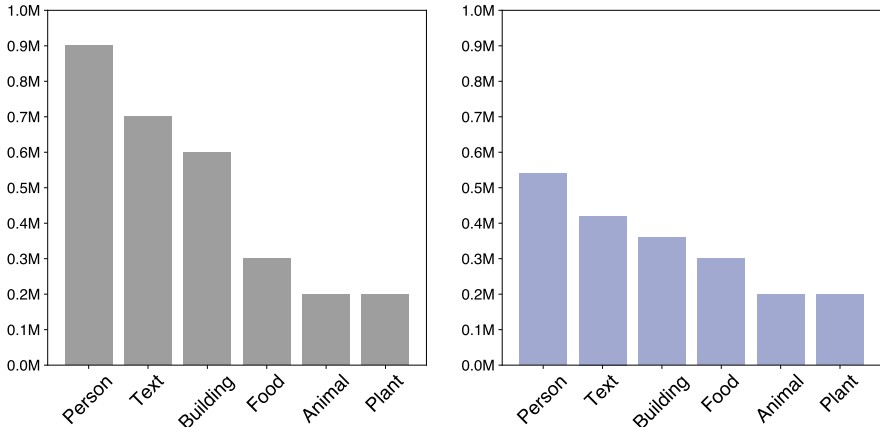

Figure 7: Top 6 objects from the subset we randomly sample from LAION-5B (left), and the distribution of the top 6 objects after simple balance (right).

## F  CASE STUDY OF PRE-TRAINING DATASET

As discussed in Section 3.2, we filter the pre-training dataset using perplexity. Figure 8 shows samples randomly selected from the top 20% and bottom 20% of the data, which have the lowest perplexity values. As illustrated, the captions in the bottom 20% of the data are more likely to contain obscure world knowledge. While augmenting the model with more world knowledge during pre-training can help it generalize better in real-world scenarios, the relatively small scale of the vision-language model pre-training dataset makes this obscure world knowledge in the bottom 20% quite sparse (seldom appearing more than once during pre-training). Consequently, this world knowledge is more likely to be noise rather than informative content for pre-training. Additionally, we also use InternVL2 (Chen et al., 2024d) to filter the pre-training dataset. The model pre-trained on the filtered subset achieves an overall score of 60.1, surpassing the model pre-trained on the original 5M dataset by 1.1 points.

## G  ABLATION ABOUT THE MAXIMUM NUMBER OF TILES

We perform more fine-grained ablation studies about the maximum number of tiles used in CATTY, and Figure 9 shows the results.

## H  ABLATION ABOUT THE FEATURE AVERAGE FROM VISION ENCODER

Before feeding features into the LLM, we compute the weighted average of features from both the general and OCR vision encoders. Figure 10 illustrates the model's performance when different weights are assigned to the general vision encoder (note that the weights assigned to the general and OCR vision encoders sum to 1). In this ablation study, we adhere to the experimental settings described in the fifth row of Table 1 in the main paper.

## I  TRAINING COST OF POINTS

We employ data parallelism (DP) (Li et al., 2020) to distribute the data and tensor parallelism (TP) (Shridhar et al., 2020) to partition the model across multiple GPUs. All our models are trained using $32 \times$ H800 80G GPUs. The pre-training stage is completed in 3 hours, while the visual instruction tuning stage takes 7 hours.

Top 20%                                             Last 20%

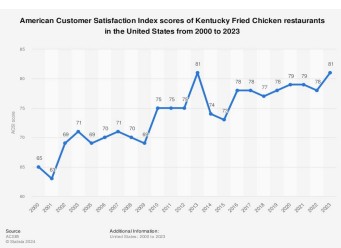

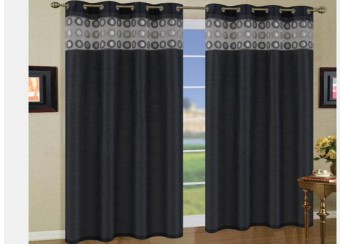

The provided line graph visually illustrates the American Customer Satisfaction Index scores of Kentucky Fried Chicken restaurants across the United States, from the year 2000 up to 2023.

The 2-Pack Natali Grommet Top Curtain Panels with Details showcase a refined aesthetic, featuring black curtains elegantly draped over a gold rod, creating a sophisticated and stylish ambiance in any room.

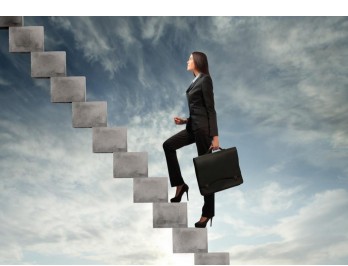

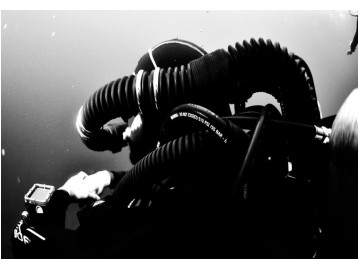

The diligent businesswoman, clad in a professional attire, steadfastly ascends the staircase, symbolizing her unwavering pursuit of goals. She carries her briefcase, embodying her dedication and commitment to her work. Each step she takes represents a strategic move towards achieving her objectives, showcasing her resilience and determination in the face of challenges.

Valéry PLATON's Shark Rebreathers Photos exhibit a striking black and white image of a diver's equipment.

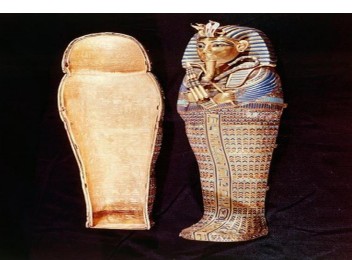

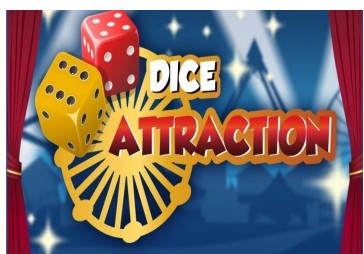

This image displays a pair of sarcophagi from Tutankhamen's tomb in the Valley of the Kings. The first sarcophagus is unadorned, while the second one is intricately crafted and engraved with gold, showcasing the pharaoh's intricate mummy wrappings. These sarcophagi represent the opulence and prestige of ancient Egyptian royalty and provide valuable insights into their burial customs and beliefs.

The Attraction Dice Online Slot Demo Game, showcased on a graphic with three dice and the words "Dice Attraction" in front of a blue background, is provided by GAMING1.

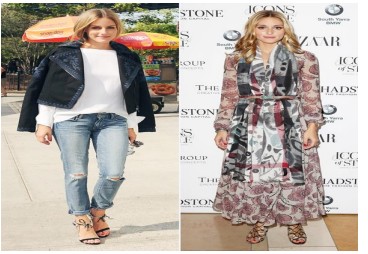

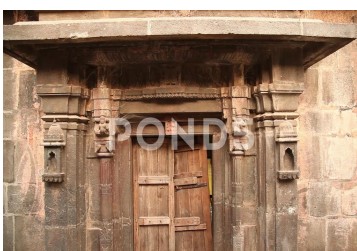

Olivia Palermo, a well-known fashion influencer, showcases her style versatility in two contrasting photos. In one, she sports a casual outfit with a comfortable, yet stylish, pair of Aquazzura shoes. In the other, she elegantly dons a more formal ensemble, making a statement with her fashion-forward choices.

The Antic Wood Door at Fort, as seen in Video Clip #72616182 on Pond5, is an old and weathered doorway featuring a wooden door and stone columns.

Figure 8: The left are samples randomly selected from the top 20% and the right are samples randomly selected from the last 20%. These obscure world knowledge is marked in red.

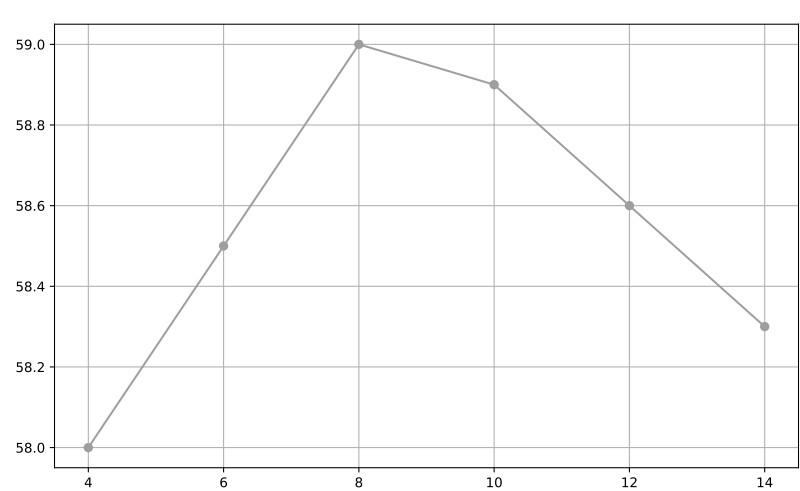

Figure 9: Ablation study about the maximum split used in CATTY.

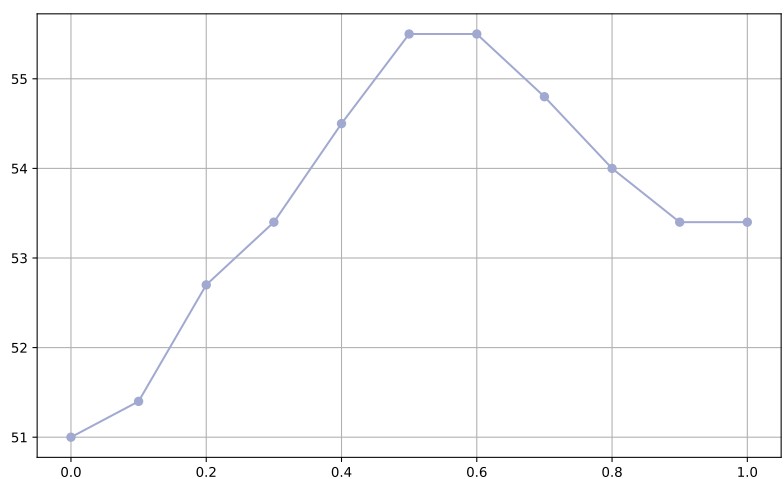

Figure 10: Ablation about the weight assigned to the general vision encoder.

