# OpenReview forum: "POINTS: Improving Your Vision-language Model with Affordable Strategies"
_ICLR.cc/2025/Conference — Submitted to ICLR 2025_

### Official Review · Reviewer_s2FU · 2024-11-01

**Soundness:** 2
**Presentation:** 3
**Contribution:** 2
**Rating:** 5
**Confidence:** 3

**Summary:**

The paper addresses the problem of model design choices and dataset selection in vision-language models learning. The authors propose a combination of existing methods to build a strong and sample-efficient baseline using a relatively small pretraining dataset of size ~1M. The main contribution is on the analysis of the dataset (individual select) and model selection (model soup) using and combining existing methods. The authors also propose a  novel image splitting strategy to tackle the well known image distortion problem in models that use a dynamic high resolution image processing.

**Strengths:**

* Well-written and easy to follow.
* Comprehensive set of VL benchmarks including hallucination and OCR benchmarks
* Table 2 shows the benefit of a perplexity-based selection of the base pretraining dataset, which is an interesting result in terms of compute efficiency.
* The authors manage to get competing performance in a wide set of VL benchmarks with a much smaller pretraining dataset (~1M)  than other methods.
* All the datasets and pretrained backbones used by the authors are publicly available which make it easily reproducible.

**Weaknesses:**

* Ablation study is thorough but the comparison with other baselines is not discussed. Specifically, POINTS is outperformed in many of the benchmarks the authors chose by similarly sized open-source models, but the differences are not discussed.
* The authors also did not discuss the choice of the base LLM (l380 Yi-1.5-9B-Chat) compared to other baselines, which makes it difficult to disentangle the reasons for the difference in performance.
* The authors introduce a QA dataset distilled form GPT-4O and include it in their base set. It is possible that this knowledge distillation alone explains why adding more data is not efficient and why a ~1M dataset is enough. The authors only justify this addition by the need to have prompt diversity but do not ablate it.

**Minor Issues**
* Small typo l480: refer to table 4 instead of table 5
* Some tables are missing the hyperlink ref (l460 and l480 for instance)
* Typo l482 “increasing adding more”
* The authors could rephrase l058 ” Unlike most existing works (Li et al., 2024; Chen et al., 2024c), we extensively ablate each newly introduced component in the strong baseline to verify their individual benefits” and not point out the lack of ablations in other works, especially when those cited work actually reference a whole blogpost about ablations in LLaVA-Next (see [1]) .
[1]  https://llava-vl.github.io/blog/2024-05-25-llava-next-ablations/

**Questions:**

* Could the authors discuss in more detail the choice and comparisons of baselines ? I acknowledge that it is easy to get lost in the number of new VLMs coming out these days but since the authors propose an analysis of the training + modeling frameworks, i would expect a comparison table that could explain the choice (and exclusion) of VLMs instances based on a few criteria (datasets, size, LLM base, ViT base, presence of OCR data, dynamic high resolution processing etc..)
* In the ablation study Table 1, what does it mean to not have IS? Does it mean training on all the potential candidates additional datasets?
* Could the authors discuss and analyze the choice of the base LLM (Yi-1.5-9B-Chat l388) ? Specifically, why not choose a base LLM that is the same as the closest baseline (for instance InternVL 1.5 that also has a dynamic high resolution processing)? It would make it easier to disentangle the contributions that come from the dataset selection, the model soup and the base LLM compared to other baselines.

Overall, the authors presented their method in a very clear and structured way. The authors also perform a thorough ablation study. However some key choices (such as GPT-4O distilled dataset, a different base LLM compared to the other baselines) are not discussed or ablated which makes it difficult to disentangle the benefit of the overall approach. The lack of discussion around the baselines and their performance makes it challenging to evaluate the novelty and impact of the proposed contribution.

---

> ### Author Response · Authors · 2024-11-17
>
> ### Disscussion about the performance of different models in Table 1
> Here are some possible reasons for the fluctuations observed in the benchmarks:
> 1. Different LLMs were used.
> 2. Different pre-training data and instruction fine-tuning data were utilized.
> 3. Different experimental parameters were applied.
>
> However, this does not affect the conclusions of our experiments or the effectiveness of our method, because during the experiments, the comparisons between our current model and the baseline were conducted with strictly controlled experimental parameters and configurations.
>
> ### Why choose Yi-1.5-9B-Chat
> 1. The main reason for using Yi-1.5-9B-Chat is that our training framework is more compatible with models configured with the LLaMA model, and at that time, we also considered overall performance, so we chose Yi-1.5-9B-Chat.
> 2. After submitting the paper, we continued to conduct experiments using Qwen2-7B, and the results are as follows.
>
> | model | MMB | MV | HB | OCR | Al2D | MMVet | MMStar | MMMU | SCI | MME | RWQ | Wild |
> | --- | --- | --- | --- | --- | --- | --- | --- | --- | --- | --- | --- | --- |
> | Yi-1. 5-9B-Chat | 83.2 | 60.7 | 48.0 | 70.6 | 78.5 | 50.0 | 56.4 | 46.9 | 92.9 | 2017.8 | 65.9 | 69.3 |
> | Qwen2-7B-Instruct | 83.2 | 62.1 | 48.2 | 71.0 | 80.3 | 52.2 | 59.6 | 49.2 | 93.8 | 2069.2 | 67.2 | 70.6 |
>
> ### GPT-4o data introduces knowledge distillation
> 1. The data generated using GPT-4o accounts for less than 1% of the total instruction fine-tuning data.
> 2. When we removed the data generated by GPT-4o, the model's performance only decreased from 61.8 to 61.7.
> Thus, the introduction of GPT-4o data does not introduce knowledge distillation
>
> ### Discussison about the different configuration of different methods
> We compare from several aspects: pre-training dataset size, instruction tuning dataset size, LLM, vision encoder, and dynamic resolution.
> | Method | LLM | Vision Encoder | PT size | SFT size | Dynamic High Resolution |
> | --- | --- | --- | --- | --- | --- |
> | Cambrian-34B | Yi-34B | CLIP ViT-L/14 | 1.2M | 7M | No |
> | Ovis1.5-LLaMA3-8B | Llama-3-8B-Ins | SigLIP-400M | 10M | 5M | DHR |
> | Idefics3-LLaMA3-8B | Llama-3-8B-Ins | SigLIP-400M | >100M | 7M | DHR |
> | InternVL2-8B | InternLM2.5-7B | InternViT-300M | - | - | DHR |
> | IXC-2.5 | InternLM2-7B | CLIP ViT-L/14 | >100M | - | DHR |
> | LLava-OneVison | Qwen2-7B | SigLIP-400M | 4.8M | 1.6M | DHR |
> | POINTS-9B | Yi-1.5-9B-Chat | CLIP ViT-L/14 | 1M | 1.2M | CATTY |
>
> ### What does it mean to not have IS?
> "Not using IS" means that we entirely used the instruction fine-tuning data from [1] and did not use Individual Select to introduce any additional data.
>
> [1] Rethinking Overlooked Aspects in Vision-Language Models

---

> ### Author Response · Authors · 2024-11-20
>
> We sincerely appreciate your great efforts in reviewing this paper. Your constructive advice and valuable comments really help improve our paper. Considering the approaching deadline, please, let us know if you have follow-up concerns. We sincerely hope you can consider our reply in your assessment, and we can further address unclear explanations and remaining concerns if any.
>
> Once more, we are appreciated for the time and effort you've dedicated to our paper.

---

> > ### Comment · Reviewer_s2FU · 2024-11-26
> > **Answer to rebuttal**
> >
> > I sincerely apologize for my delayed answer, i know it can be frustrating. To be completely honest I read your answer and the general response 3 days ago and contributions seemed less clear to me so i took the time to go over your paper again :
> >
> >  1. I get and value that your contribution is mainly on open-sourcing and detailing every step of the pipeline
> >  2. You additionally propose CATTY to handle dynamic high resolution and in order to show improved OCR performance (motivation confirmed by the competitiveness of the method in some OCR benchmark (Table 5)
> >
> > However, regarding point 1 i still feel that a lot of details are missing, i'm going to list a few here and hope it will help improve your work further :
> >  * You're using a model-soup to add additional datasets. I think that's a good idea and you show that it works. However, why not put an extensive comparison of all the baselines you compare to with the datasets they've been trained on compared to the datasets you train on, their size, their benchmark-level impact etc.. I think that would be valuable as it would inform the reader as to what is important or not for which type of benchmark.
> >  * In the same line, i got your answer about GPT-4o's dataset accounting for 1% of the overall dataset, i should be able to find that information straight away in the paper but i can't find it.
> > * The ablations are done with the overall performances but it would be more valuable to give the details about the impact of each dataset selection (something like Table 3 but detailing the gain/decrease in performance for each of the datasets and discussing it)
> > * You use a second visual encoder that is OCR-focused which induces a significant computational overhead so should be discussed and central to the analysis of the authors. For instance in Table 1 it seems that IS gives the most boost in terms of performance gain but the model without DVE and with IS is not given, so it make the CATTY contribution's advantage less clear.
> >
> > Overall i think the authors still have a lot of room for improvement, so i'm keeping my score as is!
> >
> > Do not hesitate if you have any further questions or correct me if did not get your point again, i'll do my best to be responsive! Good luck!

---

### Official Review · Reviewer_K87Y · 2024-11-03

**Soundness:** 4
**Presentation:** 3
**Contribution:** 4
**Rating:** 5
**Confidence:** 4

**Summary:**

The paper proposes to improve the baseline LLaVA model in terms of the following three aspects: (a) to refine the pre-training dataset quality, the paper introduces CapFusion utilizing an LLM to integrate world knowledge from existing captions while simplifying their structures, (b) the model architecture is improved through the introduction of dynamic high resolution component with consistent aspect ratio, and (c) the visual instruction tuning is enhanced through the usage of model soup that merges weights from models fine-tuned on different learning rates.

**Strengths:**

- The proposed components are indeed effective and target the limitation of existing vision-language models in great detail.
- Compared to finetuning dataset selection, the proposed model soup is a more sustainable design choice in terms of the return on performance with scale.
- Perplexity-based dataset selection technique is simple and could have a potentially broader impact on data-efficient pretraining of large foundation models.

**Weaknesses:**

- While the paper is overall well-written, it is hard to spot the motivation behind the introduced components now and then. For instance, the introduction states "... However, recent advancements have rendered its (LLaVa's) performance suboptimal. Thus,
 there is a need to establish a stronger baseline for further exploration..." Without proper references or an explicit description of how LLaVa's performance has gone suboptimal, the current motivation reads as "We know the model x and we want to improve upon it." While the latter motivation is indeed genuine, this however opens up a plethora of search space where the model x can be improved. As such, I am reluctant about why the paper would then improve model x in only a few chosen directions.

- Continuing on the above note, a good part of the experiments are occupied with performance on OCR-related tasks. However, there is no such concrete discussion of why the paper chooses to highlight the OCR task, what limitations the current baselines have for the OCR task, and what exactly are the example scenarios where the proposed model performs better on OCR than the baseline LLAVA.

- My major concern with the current experiments is a lack of discussion on the additional resource overhead brought about by the proposed components. For a practitioner to prefer the proposed POINTS over LLAVA, it would be helpful if the paper mentions: (a) the overhead in pretraining/finetuning in terms of computational resources such as GPU memory, training/inference times in terms of wall-clock/FLOPs as well as the data efficiency of the proposed method. On the lattermost note, I would like to see a discussion on how the POINT model performs compared to LLAVA if a downstream fine-tuning task has limited (few-shot) training examples.

- How does the proposed CATTY method scale with the image size in terms of the model inference time?

Minor suggestions:

- Introduce the full name for OCR before the first usage of the acronym in the paper (line 057).
- Lines 75-78: citation required for ".. Compared to conducting model soup on models fine-tuned with different hyper-parameters, e.g. learning rate, the improvement with model soup on models fine-tuned with different datasets is much more prominent."

**Questions:**

See Weaknesses. Most of my current concerns with the paper are around a lack of proper motivation and analyses of the proposed components.

---

> ### Author Response · Authors · 2024-11-17
>
> ### Motivation is unclear
> Our motivation is mainly divided into three points:
>
> 1. The leading models in the industry generally do not disclose their training details.
> 2. There has been no in-depth exploration of how pre-training data is processed in current open-source work.
> 3. During the instruction fine-tuning phase, most current work focuses on continuously adding new data, which becomes difficult to improve upon in the later stages.
>
> In response to the above motivations, we further propose:
>
> 1. Building a stronger baseline for subsequent experiments: A stronger baseline often determines the final performance of the model. Many previous works have conducted experiments based on LLaVA, but with the development over the past year, many new improvements have emerged in this field, such as DHR and CapFusion. Using existing methods, we have obtained a stronger baseline based on LLaVA.
> 2. For processing pre-training data, we propose using perplexity to filter the data, thereby achieving better performance with less data.
> 3. During the instruction fine-tuning phase, to address the bottleneck encountered with newly added datasets, we propose using model soup with models trained on different data, which can further enhance the model's performance.
>
> ### Why highlight OCR-related tasks?
> 1. OCR is a very important direction for MLLM applications. Many previous works have focused on enhancing the OCR performance of multimodal models, such as Vary[1]. However, enhancing OCR capability is not our primary goal. In our entire work, only the Dual Vision Encoder and CATTY are directly related to OCR. Our main objective is to improve the overall capability of the model, with enhancing OCR-related abilities being just one aspect.
> 2. LLaVA-Next's capability in OCR is still quite limited, as can be seen from its performance on OCRBench. LLaVA-Next scores 53.1 (according to OpenCompass results), while our model scores 70.6. In terms of approach, LLaVA: 1) did not significantly enhance the OCR capability of the vision encoder, and 2) applied padding to images in the use of DHR, which might affect the model's learning. To address these issues: 1) we introduced an additional OCR-enhanced vision encoder, and 2) we proposed CATTY, which segments images without padding and while maintaining the aspect ratio.
>
> ### No mention of computation overhead
> In Appendix I, we explain our training costs
>
> ### How does POINTS perform when the downstream test tasks have very few training samples?
> Additionally, we further tested on the recently released MEGABench, which covers over 500 real-world tasks, most of which do not have related samples in the training set. The specific results are as follows:
> | Method | Overall SI |
> | --- | --- |
> | InternVL2-8B | 29.21 |
> | POINTS-Qwen2-7B | 26.10 |
> | MiniCPM-V2.6-8B | 25.95 |
> | LLava-OneVision-7B | 25.69 |
> | Idefics3-8B-LLaMA3 | 12.06 |
> It can be observed that POINTS still exhibits good generalization performance.
>
> ### How does the inference time of CATTY change as the image size increases?
> CATTY and DHR have similar scaling curves for image size versus inference time.
> $$
> \mathrm{I n f e r e n c e T i m e} \propto \frac { H } { w } \times \frac { W } { w } \times w ^ { 4 } = H W w ^ { 2 }
> $$
>
> Here, H and W are the height and width of the image, and w is the window size.
>
> ### Lines 75-78 misses citation
> This is the phenomenon we observed in our experiments. Please refer to the second row of Table 4 for the experiment.
>
>
> [1] Vary: Scaling up the Vision Vocabulary for Large Vision-Language Models

---

> > ### Comment · Reviewer_K87Y · 2024-11-19
> >
> > Thank you for your response and additional experiments.
> >
> > Per your rebuttal, an obvious question would be - why not structure the paper writing (title, abstract, introduction) to be centered around the task of OCR then? Since I see that a large chunk of the experiments and design choices being devoted to the OCR task, why not just make it more obvious for the readers? For instance, the current title suggests me to look for contributions in the paper that -- indicated by the keyword "affordable" -- increase the resource/data efficiency of Vision-Language models across a range of more general tasks/setups. Can the authors comment on this?
> >
> > Also, can the authors report the wall clock time of CATTY with varying image sizes?

---

> ### Author Response · Authors · 2024-11-20
>
> ### Why not structure the paper to be centered around OCR?
> 1. Thank you for your suggestion. But as we mentioned earlier, our main goal is to achieve performance comparable to industry-leading models with fewer resources. Our focus is on the general performance of the model, and OCR is just one aspect of it. In this work, most of our efforts are directed towards general performance, not just OCR. For example, we use perplexity to filter pre-training data and employ model soup.
> 2. ``Affordable`` means that all the methods adopted in this paper are quite lightweight, allowing us to achieve a model comparable to the state-of-the-art (SOTA) with fewer resources. For instance, our pre-training data consists of only 1 million samples, and using 32 x H800 GPUs, we can complete all model training within 10 hours (please refer to Appendix I). This approach is a good option for teams with limited resources.
>
> ### Wall clock time of CATTY with varying image sizes?
> We chose the following experimental configuration to test the wall clock time of CATTY and DHR:
> ```
> Device: 1 x 80G H800
> Model: POINTS-Qwen2-7B
> Inference framework: HuggingFace transformers
> Time calculation method: Measure the forward time for 10 images using teacher forcing mode.
> ```
> The images we use are square-shaped, and the content fed into the language model includes an image and its caption. To highlight the impact of image sequence length on wall clock time, we have chosen relatively short captions (10 tokens).
>
> | Method/Resolution | 384   | 512   | 640   | 768   | 896   | 1024  | 1280  |
> |-------------------|-------|-------|-------|-------|-------|-------|-------|
> | CATTY             | 0.373s| 0.375s| 0.631s| 0.632s| 0.644s| 0.647s| 0.673s|
> | DHR               | 0.367s| 0.375s| 0.620s| 0.629s| 0.646s| 0.647s| 0.668s|

---

### Official Review · Reviewer_QdCF · 2024-11-03

**Soundness:** 2
**Presentation:** 3
**Contribution:** 2
**Rating:** 5
**Confidence:** 4

**Summary:**

The paper proposes affordable and efficient methods to improve vision-language models, focusing on three main parts: (1) building a robust baseline model with thorough validation, (2) using perplexity filtering to improve the quality of pre-training data, and (3) applying "model soup" to merge weights from different fine-tuned models. The final model, POINTS, achieves competitive performance with state-of-the-art models while remaining lightweight and accessible. The authors also introduce improvements like CATTY to reduce image distortion and enhance OCR capabilities, all while keeping computational costs manageable.

**Strengths:**

- The use of perplexity-based filtering of pre-training data is inspired by its success in large language models. This approach is a creative application of an existing concept in a new context, addressing data quality for vision-language models.
- The paper is well-structured, with clear divisions between each method's presentation and the experimental results validating those methods.
- The paper's focus on affordable strategies is practical for the broader community, especially for those without extensive computational resources. Achieving competitive performance using lightweight techniques is useful.

**Weaknesses:**

- Each of the proposed methods, such as perplexity filtering, CATTY, and model soup, are either existing methods or slight modifications of existing approaches. This limits the overall novelty of the contributions. More significant deviations or novel techniques could make the paper's contributions more impactful.
- While the authors claimed model soup has introduced improvements, the paper does not thoroughly explore its potential limitations or generalizability across different tasks or datasets. There are only limited details on the possible negative effects of merging model weights, particularly if the datasets have significant divergence in distribution. Also, I wonder if there can be potential overfitting to benchmark from the Greedy Soup method and why "MathV360K" has no improvement while others have improved.
- While perplexity filtering effectively reduces the dataset size while preserving model performance, there is a potential risk that the model's ability to generalize could regress due to a lack of exposure to a diverse range of data. The paper does not provide any experiments or discussion addressing this potential drawback. I think it is important to understand the gap in the understanding of the impact of data reduction on generalization performance.

**Questions:**

- While perplexity filtering effectively reduces the dataset size while preserving model performance, there is a potential risk that the model's generalization ability could regress due to reduced exposure to diverse data. Could you provide more insights or experimental evidence on the impact of this data reduction on the model's generalization capabilities?
- Regarding the Model Soup technique, is there a potential risk of overfitting to benchmarks due to the greedy soup method? Could you elaborate on why "MathV360K" showed no improvement while other datasets did?
- Have you observed any cases where CATTY underperforms compared to DHR? It would be useful to understand any trade-offs involved, particularly in terms of applicability to different image types.
- Typo in the table 4 "Greey Soup" -> "Greedy Soup"

**Details Of Ethics Concerns:**

There are no ethical concerns in the paper.

---

> ### Author Response · Authors · 2024-11-17
>
> ### Potential limitations of model soup
> 1. Currently, we have observed that although there are significant gains in the early stages, it becomes increasingly difficult to improve the model's performance as the process continues. This phenomenon has also been noted in a recent study[1]. The main reason is that as model soup progresses, the similarity between models increases, making it challenging for the model to break through the original local optimum to reach a better local optimum.
> 2. We believe that the distribution differences in the dataset do not have a significant negative impact on model soup; instead, they may actually provide benefits. For example, as shown in Figure 3 of the original paper, performing model soup with a model trained on the Base Set + MathV360K and a model trained on the Base Set + ScreenQA can significantly improve the model's performance. MathV360K is a dataset related to mathematics, while ScreenQA is a dataset related to screen-based question answering. There are significant differences between the two datasets in terms of both images and questions.
> 3. Model soup do not show overfit to a specific benchmark. We conducted model soup primarily based on the Overall Score metric in Table 1. However, in the final presentation of results, we also provided metrics for MME, RealWorldQA, and LLaVA-Wild, where we achieved good performance as well. Additionally, we further tested on the recently released MEGABench and obtained impressive results. The specific results are shown below.
>
> ### Risks of reduced generalization ability due to a smaller pre-training dataset size
> 1. When selecting model data, we primarily referred to the Overall Score metric in Table 1. However, in the final presentation of results, we also provided metrics for MME, RealWorldQA, and LLaVA-Wild, on which we achieved good performance as well.
> 2. Additionally, we further tested on the recently released MEGABench[2], which covers over 500 real-world tasks. The specific results are as follows:
> | Method | Overall SI |
> | --- | --- |
> | InternVL2-8B | 29.21 |
> | POINTS-Qwen2-7B | 26.10 |
> | MiniCPM-V2.6-8B | 25.95 |
> | LLava-OneVision-7B | 25.69 |
> | Idefics3-8B-LLaMA3 | 12.06 |
> To maintain consistency with previous works, such as LLaVA-OneVision, we have newly added the POINTS-Qwen2-7B model, which uses Qwen2-7B as the LLM.
> 3. The current pre-training of MLLMs is essentially a process of aligning language and image signals. The learning of knowledge has already been accomplished separately during the training of the Vision Encoder and the LLM. Therefore, using less data during the alignment phase does not significantly impact the model's ability to acquire knowledge.
>
> In summary, the reduction in pre-training data has not weakened the generalization capability of POINTS.
>
> ### Have any areas been observed where CATTY is not as good as DHR?
> 1. In our response to the Reviewer eeM9, we mentioned that CATTY introduces additional computational overhead compared to DHR, but this overhead is negligible relative to the total computational cost.
> 2. As shown in Table 1, the main advantage of CATTY over DHR is its improved performance on OCR-related tasks, such as OCRBench. This is because OCR tasks are more sensitive to image pixels, a point that has been noted in previous work[3]. However, for some other tasks, the performance of CATTY is comparable to, or even slightly lower than, that of DHR, such as in MMVet. But compared to DHR, CATTY will bring overall improvement to the model.
>
> [1] Exploring Model Kinship for Merging Large Language Models
>
> [2] MEGA-Bench: Scaling Multimodal Evaluation to over 500 Real-World Tasks
>
> [3] Improved Baselines with Visual Instruction Tuning

---

> ### Author Response · Authors · 2024-11-20
>
> We sincerely appreciate your great efforts in reviewing this paper. Your constructive advice and valuable comments really help improve our paper. Considering the approaching deadline, please, let us know if you have follow-up concerns. We sincerely hope you can consider our reply in your assessment, and we can further address unclear explanations and remaining concerns if any.
>
> Once more, we are appreciated for the time and effort you've dedicated to our paper.

---

### Official Review · Reviewer_eeM9 · 2024-11-03

**Soundness:** 3
**Presentation:** 3
**Contribution:** 2
**Rating:** 3
**Confidence:** 4

**Summary:**

This paper proposes a robust baseline vision-language model. The main contributions are threefold: (1) Framework improvement: the authors introduce Consistent Aspect Ratio Dynamic High Resolution for the vision encoder of the multimodal LLM; (2) Data filtering: the authors use perplexity to filter the pre-training dataset; (3) Model ensemble: this work employs model soup methods to merge models fine-tuned with different datasets to achieve better results.

**Strengths:**

- Paper is clearly written.

- Introduced approaches are technically sound.

**Weaknesses:**

- Novelty concern: The proposed strong baseline model primarily integrates multiple existing advancements in vision-language models, with limited novel technical contributions or findings. Specifically, the proposed CATTY encoding combines dynamic high-resolution encoding and sliding window techniques, both of which have been explored in previous works [1-3]. Additionally, model soup or model ensemble techniques have been shown to effectively boost performance. The novelty of the introduced maximum, average, and greedy soup methods for merging model results is limited.

- Incomplete experimental analysis: Experimental results comparing CATTY with DHR are shown in Table 1. However, the proposed CATTY encoding approach, which leverages a sliding window with overlapping, may also introduce extra computation costs, additional vision tokens, or repetitive information. This analysis is crucial for understanding the pros and cons of using CATTY for future work, yet these aspects are not examined.

- Confusing table and figure references: In the experiments section, some table and figure references are confusing. For instance, in Line 480, should this refer to Table 3? And in Line 484, should this be Table 4? This paper may require more attention to detail in the writing.

[1] Chen, Zhe, et al. "How far are we to gpt-4v? closing the gap to commercial multimodal models with open-source suites." arXiv preprint arXiv:2404.16821 (2024).

[2] Lin, Ziyi, et al. "Sphinx: The joint mixing of weights, tasks, and visual embeddings for multi-modal large language models." arXiv preprint arXiv:2311.07575 (2023).

[3] Liu, Ze, et al. "Swin transformer: Hierarchical vision transformer using shifted windows." Proceedings of the IEEE/CVF international conference on computer vision. 2021.

**Questions:**

- Is the baseline in Table 4 trained only on a base instruction tuning dataset, or is it trained on a combination of a base instruction tuning dataset and a series of visual instruction tuning datasets? How was the baseline model trained?

- What specific off-the-shelf vision-language model was used to calculate perplexity in Table 2? In the experiments, did you observe any differences when using different VLMs?

---

> ### Author Response · Authors · 2024-11-17
>
> ### Missing analysis about CATTY
> 1. For the 1M pre-training data used, we calculated the number of tokens obtained using CATTY and DHR respectively. Compared to DHR, CATTY introduced approximately 10% more original vision tokens. This newly introduced number of tokens accounts for about 2% of the total number of tokens, so it essentially does not introduce significant computational overhead.
> 2. Regarding redundant information, we conducted statistics on the 1M pre-training data and found that the average overlap between every two adjacent windows is 10 pixels. Intuitively, we can infer that redundant information might affect the model's counting ability. Therefore, we selected two specific metrics: ``Object Localization`` from MMBench (which includes counting problems) and ``Count`` from MME. The performance of CATTY and DHR is as follows:
> | Method  | Object Localization (MMBench) | Count (MME) |
> |-------|-------------------------------|-------------|
> | DHR   | 77.1                          | 168.3       |
> | CATTY | 77.2                          | 167.8       |
> 3. Since CATTY introduced an additional 10% of vision tokens, and previous work, such as Idefics2[1], has shown that additional vision tokens can enhance model performance, we further enlarged the width and height of the images by a factor of 1.05 (sqrt(1.1)) before using DHR. We denote this set of experiments as DHR(large). The specific experimental results are as follows:
> | Method | Overall score |
> | --- | --- |
> | CATTY | 53.4 |
> | DHR | 52.6 |
> | DHR (large) | 52.4 |
> ### How the baseline in Table 4 is obtained
> The baseline in Table 4 is derived from the last row of Table 1, which utilizes the Base Set discussed in the **Individual Selection** section of 3.1. Further details about the Base Set are provided in Table 6 (marked in black) in the appendix.
> ### VLM used in pre-training dataset filtering
> Currently, the model we use for PPL filtering is the one obtained from the last experiment in Table 1. Additionally, we used another VLM (InternVL2-8B) to filter the data (as detailed in Appendix F). Pre-training on the data filtered by InternVL2-8B resulted in an overall score of 60.1.
>
> [1] What matters when building vision-language models?

---

> ### Author Response · Authors · 2024-11-20
>
> We sincerely appreciate your great efforts in reviewing this paper. Your constructive advice and valuable comments really help improve our paper. Considering the approaching deadline, please, let us know if you have follow-up concerns. We sincerely hope you can consider our reply in your assessment, and we can further address unclear explanations and remaining concerns if any.
>
> Once more, we are appreciated for the time and effort you've dedicated to our paper.

---

### Author Response · Authors · 2024-11-17
**General Response**

We greatly appreciate the valuable suggestions from the reviewers. Here, we first address the common issues raised by the reviewers, and then we address each reviewer's individual concerns one by one.

### Novelty of this paper
Regarding the novelty of our paper, we would like to further elaborate on our perspective:
1. As we mentioned at the beginning of the article, our primary goal is not to propose new methods, but to refine the existing model training procedures to build a multimodal model that can rival industry-leading models in three key aspects: i) constructing a stronger baseline for subsequent experiments, ii) processing pre-training data, and iii) enhancing the model's performance during the instruction fine-tuning phase by not only continuously introducing more datasets but also utilizing model soup.
2. It is well known that most of the current industry-leading models, such as Qwen2-VL and InternVL2, rarely explain the details of their model training to the community. This practice makes it difficult for others to train a model from scratch that can rival the state-of-the-art (SOTA), which is detrimental to the progress of the field. The most significant contribution of POINTS is that it provides a detailed explanation of every step in the model training process, from data collection and processing to model architecture and training configuration. We hope that this approach can offer valuable references to our peers in the community.
3. In the era of large models, one should not overly rely on the innovation of methods themselves to judge the contribution of a work. Instead, a more comprehensive and detailed examination of every step in the model construction process should be conducted to unleash the model's potential performance. Although POINTS utilizes many existing methods, compared to previous approaches such as LLaVA-Next, we have meticulously fine-tuned every step of the training process.

### Typos in this manuscrip
ased on the reviewer's feedback, we have corrected the typos and updated the manuscript. Additionally, for the reviewer's convenience, we have included the appendix at the end of the main text (previously, the appendix was placed in a zip file).

---

### Meta-Review · Area_Chair_DQgT · 2024-12-18

**Metareview:**

This paper discusses critical design choices when training LLaVA-like vision-language models. The authors explore three main components : the training data, image splitting, and model averaging. All three components seem to provide improvements. Reviewers have noted that the paper lacks details and could be improved in a variety of ways. While defining strong and simple baselines is of outstanding value to the community, the current form of this work is not good enough to warrant acceptance at ICLR.

**Additional Comments On Reviewer Discussion:**

The authors have submitted a rebuttal addressing reviewers comments. Two out of four reviewers have answered back, but judging that the rebuttal content did not change their assessment of the quality of this work - keeping a rejection rating.

---

### Decision · Program_Chairs · 2025-01-22

Reject